# Heparan-6-*O*-Endosulfatase 2 Promotes Invasiveness of Head and Neck Squamous Carcinoma Cell Lines in Co-Cultures with Cancer-Associated Fibroblasts

**DOI:** 10.3390/cancers15215168

**Published:** 2023-10-27

**Authors:** Pritha Mukherjee, Xin Zhou, Julius Benicky, Aswini Panigrahi, Reem Aljuhani, Jian Liu, Laurie Ailles, Vitor H. Pomin, Zhangjie Wang, Radoslav Goldman

**Affiliations:** 1Department of Oncology, Lombardi Comprehensive Cancer Center, Georgetown University, Washington, DC 20057, USA; pm1226@georgetown.edu (P.M.); xz42@georgetown.edu (X.Z.); jb2304@georgetown.edu (J.B.); aswini.panigrahi@georgetown.edu (A.P.); 2Biotechnology Program, Northern Virginia Community College, Manassas, VA 20109, USA; 3Clinical and Translational Glycoscience Research Center, Georgetown University, Washington, DC 20057, USA; ra1091@georgetown.edu; 4Department of Biochemistry and Molecular & Cellular Biology, Georgetown University, Washington, DC 20057, USA; 5Division of Chemical Biology and Medicinal Chemistry, Eshelman School of Pharmacy, University of North Carolina, Chapel Hill, NC 27599, USA; jianliu@glycantherapeutics.com; 6Department of Medical Biophysics, University of Toronto, Toronto, ON M5G 1L7, Canada; laurie.ailles@uhnresearch.ca; 7Princess Margaret Cancer Centre, University Health Network, Toronto, ON M5G 1L7, Canada; 8Department of BioMolecular Sciences, University of Mississippi, Oxford, MS 38677, USA; vpomin@olemiss.edu; 9Research Institute of Pharmaceutical Sciences, School of Pharmacy, University of Mississippi, Oxford, MS 38677, USA; 10Glycan Therapeutics, LLC, 617 Hutton Street, Raleigh, NC 27606, USA; zhangjie.wang@glycantherapeutics.com

**Keywords:** head and neck squamous cell carcinoma, cancer-associated fibroblast, heparan-6-*O*-endosulfatase, cancer invasion, Sulf-2 CRISPRCas9 knockout, Sulf-2 inhibitor

## Abstract

**Simple Summary:**

Local invasion of cancer cells is an early step in the cascade of metastasis that requires cooperation of multiple factors and cell types in the tumor microenvironment (TME). One important factor is the crosstalk of cancer cells with cancer-associated fibroblasts (CAFs). Here we explore the impact of a secretory enzyme, heparan-6-*O*-endosulfatase 2 (Sulf-2), on the CAF-assisted invasion of head and neck squamous carcinoma cells into Matrigel. We show that Sulf-2 knockout inhibits cancer cell invasion in a spheroid co-culture model and we identified a novel Sulf-2 inhibitor that, in the same model, limits cancer cell invasion.

**Abstract:**

Local invasiveness of head and neck squamous cell carcinoma (HNSCC) is a complex phenomenon supported by interaction of the cancer cells with the tumor microenvironment (TME). We and others have shown that cancer-associated fibroblasts (CAFs) are a component of the TME that can promote local invasion in HNSCC and other cancers. Here we report that the secretory enzyme heparan-6-*O*-endosulfatase 2 (Sulf-2) directly affects the CAF-supported invasion of the HNSCC cell lines SCC35 and Cal33 into Matrigel. The Sulf-2 knockout (KO) cells differ from their wild type counterparts in their spheroid growth and formation, and the Sulf-2-KO leads to decreased invasion in a spheroid co-culture model with the CAF. Next, we investigated whether a fucosylated chondroitin sulfate isolated from the sea cucumber *Holothuria floridana* (HfFucCS) affects the activity of the Sulf-2 enzyme. Our results show that HfFucCS not only efficiently inhibits the Sulf-2 enzymatic activity but, like the Sulf-2 knockout, inhibits Matrigel invasion of SCC35 and Cal33 cells co-cultured with primary HNSCC CAF. These findings suggest that the heparan-6-*O*-endosulfatases regulate local invasion and could be therapeutically targeted with the inhibitory activity of a marine glycosaminoglycan.

## 1. Introduction

Head and neck squamous cell carcinoma (HNSCC) is the most common type of head and neck cancer. It is an epithelial cancer, arising from the mucosa of the upper aerodigestive tract [1,2]. During the process of carcinogenesis, accumulation of multiple genetic insults imparts invasive capabilities to the tumor. The HNSCC cells first invade the basement membrane of the native epithelium before proceeding to lymph node metastasis in approximately 50% of the patients; positive nodes are associated with a substantially decreased survival [1,3,4]. It is important to understand the features and mediators of the local HNSCC invasion so that new diagnostic methods and treatment approaches can be developed to prevent the spread of the disease.

Tumor cell invasion and metastasis is of primary importance in the prognosis of cancer patients [5,6]. Local invasion is an early step in the cascade and the penetration of cancer cells into the neighboring tissues requires adhesion, proteolysis of the extra cellular matrix (ECM) components, and migration of the cancer cells [5,7,8]. Co-operation of the cancer cells with their microenvironment can promote the local invasion and specific types of CAFs are known to assist with this process [9,10]. It is already established that CAFs modulate cancer cell invasion directly by several factors, including secretion of pro-invasive stimuli, remodeling the ECM, and tumor–stroma crosstalk. Apart from the secretory factors, the physical interaction of CAFs with cancer cells is required for the contraction and alignment of ECM that facilitates local invasion [11,12]. CAFs have been denoted as significant tumor-supporters in HNSCC. They promote tumor cell proliferation and angiogenesis, and are strongly correlated to invasion, treatment resistance, and poor patient outcome in HNSCC. CAFs exert their effect on tumor progression through releasing various growth factors (EGF, VEGF, HGF, TGF-β, PDGF-A), cytokines (IL-6), and chemokines (CXCL8). In the TME, CAFs represent the primary source of matrix metalloproteases (MMPs) which remodel the ECM, releasing active matrix-embedded growth factors like FGFs and TGF-β, further promoting tumor invasion [13,14].

Heparan sulfate proteoglycans (HSPGs), ubiquitous components of cell surfaces and of the ECM, alter the progression of cancers through multiple mechanisms [15,16]. HSPGs act as ligands or co-receptors in the signaling cascades of the FGF, VEGF, Wnt, Notch, IL-6/JAK-STAT3, or NF-κB pathways. Because of this property, they modulate the expression of cytokines, chemokines, growth factors, and adhesion molecules and, at the same time, regulate their gradients in the ECM. The heparan-6-*O*-endosulfatases Sulf-1 and Sulf-2 are secreted enzymes that post-synthetically modulate the sulfation of heparan sulfate (HS) chains. Their activity creates structural variability in the sulfation of HSPG domains [17]. The degree and patterns of HS sulfation gives rise to highly negatively charged regions promoting strong ionic interactions with positively charged amino acid residues of binding partners. The post-synthetic editing of the HS by Sulf-1 and Sulf-2 dictates the binding affinity of this glycosaminoglycan (GAG) to many proteins including transmembrane receptors, ECM structural proteins, and secreted signaling molecules [18,19], which modulates their distribution and function.

The expressions of Sulf-1 and Sulf-2 is dysregulated in many cancers. Earlier studies correlated high levels of endogenous Sulf-1 and Sulf-2 with survival and tumorigenesis of hepatocellular carcinoma [20], pancreatic carcinoma [21], gastric carcinoma [22], invasive breast carcinoma [23], cervical carcinoma [24], urothelial carcinoma [25], glioblastoma [26], as well as multiple myeloma [27]. We have shown that the Sulfs are upregulated in many cancers and negatively affect survival of HNSCC patients in a stage-specific manner [28]. Sulf-1 is supplied to the tumors primarily by the CAF, while Sulf-2 is more abundant in the epithelial HNSCC tumor cells [28,29,30]. The widespread upregulation of Sulfs in cancer and their impact on survival suggest that they should be explored as diagnostic and therapeutic targets [31].

In this study, we focused on the contribution of Sulf-2, dominant in the cancer cells, to the local invasiveness of HNSCC cell lines in a spheroid co-culture model with primary HNSCC CAF. We generated Sulf-2-deficient HNSCC cell lines via gene editing using CRISPR/Cas9, and their phenotypes were compared to wild-types. Furthermore, we assessed the effect of a novel inhibitor, a unique marine GAG named HfFucCS, on Sulf-2 activity. Our results show that Sulf-2 affects the growth and invasion of HNSCC spheroids and we demonstrated that HfFucCS inhibits not only the Sulf-2 enzymatic activity but also cancer cell invasion in the spheroid model. Our results warrant further exploration of the heparan-6-*O*-endosulfatases in HNSCC and other malignancies.

## 2. Materials and Methods

### 2.1. Materials

Cell culture media (DMEM/F12, IMDM) and Growth Factor Reduced Matrigel were from Corning Inc., Corning, NY, USA. Lenti-X HEK293T cells were from Takara Bio, San Jose, CA, USA. Vectors and plasmids were procured from Addgene (Watertown, MA, USA): pHR-CMV-TetO2-3C-TwinStrep-IRES-EmGFP vector (plasmid # 113885), envelope plasmid (pMD2.G, plasmid # 12259), and packaging plasmid (psPAX2, plasmid # 12260). Gene Knockout Kit v2 and Cas9 nuclease were from Synthego, Redwood City, CA, USA. Nucleofection Kit V Complete Solution and Nucleofector 2 were from Lonza (Basel, Switzerland). Phusion Plus PCR Master Mix was from Thermo Fisher Scientific, Waltham, MA, USA. Hydrocortisone was from MilliporeSigma, Burlington, MA, USA. Cell culture supplements were from Gibco^TM^, Billings, MT, USA. Cell culture flasks and dishes were from Nunc, Thermo Fisher Scientific, Rochester, NY, USA; 96-well round bottom low attachment plates and 384-well black plates were from Corning, Corning, NY, USA. Polyethylenimine was from Polysciences, Warrington, PA, USA. Polybrene was from Santa Cruz Biotechnology, Santa Cruz, CA, USA. 4-Methylumbelliferyl sulfate (4-MUS) substrate was from Sigma-Aldrich (St. Louis, MO, USA) and GlcNS6S-GlcA-Glc6SNS-IdoA2S-GlcNS6S-IdoA2S-GlcNS6S-GlcA-pNP (2S2-6S4) substrate was from Glycan Therapeutics, Raleigh, NC, USA. Heparan sulfate from porcine mucosa (GAG-HS01) was obtained from Iduron, Alderley Edge, UK.

### 2.2. Cell Culture

The cell lines SCC35 and Cal33 are human squamous cell carcinoma cell lines derived from a hypopharynx tumor and the oral cavity, respectively. The SCC35 cell line was kindly provided by Prof. Vicente Notario, Georgetown University, Washington, DC, USA. CAL33 were purchased from DSMZ (Leibniz-Institute DSMZ, Braunschweig, Germany). Primary head and neck fibroblasts, HNCAF-61137 (HNCAF37), were derived at the Princess Margaret Cancer Centre, Toronto, CA from a post-surgery tumor sample of a patient (male/59 years of age) before radio/chemo-therapy in line with an approved Research Ethics Board protocol. The tumor site was the tongue and it was categorized as SCC, stage IVA, T3/N2b/M0. Samples were dissociated into single cell suspensions and plated in 75 cm^2^ flasks in IMDM + 10% FBS + 1% pen/strep at a high density and cultured a in 5% CO_2_ atmosphere. The medium was changed several days after plating and the CAF line was allowed to expand while the tumor cells died out after several passages. The line was tested for alpha-smooth muscle actin expression and was confirmed negative for pan cytokeratin as previously described [32]. SCC35 cells were grown in DMEM/F12 supplemented with 400 ng/mL hydrocortisone. Cal33 and HNCAF37 cells were grown in IMDM. All media were supplemented with 10% fetal bovine serum, non-essential amino acids and 1 mM sodium pyruvate, and were grown in a humidified 5% CO_2_ atmosphere. Cells were sub-cultured at 90–100% confluence. The HNCAF37 cells were slower in growth compared to the cancer cells and needed passaging every 5 days at 1:3 when they reached 90–100% confluence.

### 2.3. Generation of Fluorescently Labeled Cells

GFP-expressing SCC35 cells were generated via lentiviral transduction of pHR-CMV-TetO2-3C-TwinStrep-IRES-EmGFP vector. Briefly, lentiviral particles were generated via co-transfection of the above transfer vector with envelope (pMD2.G) and packaging (psPAX2) plasmids to Lenti-X HEK293T cells using linear 25 kDa polyethylenimine at a ratio of 30 µg DNA to 75 µg PEI in 75 cm^2^ flasks in DMEM/F12 containing 2% FBS. The conditioned media containing lentiviral particles were collected 3 days post-transfection, filtered through 0.45 µm PES filter, mixed 2:1 with SCC-35 culture medium, and supplemented with 1 µg/mL polybrene. The above transduction mix was applied to SCC-35 cells for 48 h before medium exchange and subculture. GFP-labeled Cal33 cells were generated via lentiviral transduction as described [33].

### 2.4. Generation of CRISPR/Cas9 Sulf-2 Knockout Cells

Sulf-2-deficient SCC35 and Cal33 cell lines were generated using CRISPR/Cas9 gene editing using the Sulf-2-specific Gene Knockout Kit v2 as described previously [34]. Briefly, ribonucleoprotein complexes consisting of 180 pmoles of Sulf-2-targeting synthetic sgRNAs and 20 pmoles of S. pyogenes Cas9 nuclease were nucleofected into 5 × 10^5^ SCC35 or Cal33 cells in Nucleofection Kit V Complete Solution, using Nucleofector 2 and the X-005 program according to the manufacturer’s instructions, and subsequently subjected to clonal expansion to select clones with high knockout efficiencies. The knockout efficiency was determined using Inference of CRISP Edits (ICE) analysis as described [34]. Briefly, the targeted region of genomic DNA from CRISPR/Cas9-edited and wild-type cells was amplified using PCR using Phusion Plus PCR Master Mix, 100 ng of genomic DNA, and 0.5 µM of each primer, 5′ GACCCAGTTGTGACCCCAAA 3′ (forward) and 5′ AGTGAAGAGGGACGAGGGAA 3′ (reverse). The amplified DNA was submitted to Sanger sequencing (Genewiz, Chelmsford, MA, USA) using the sequencing primer 5′ GGCCCGTGTCTTGGTTTCAATGTAC 3′. Trace files were analyzed by ICE Analysis Tool (Synthego, https://ice.synthego.com). Knockout of the enzyme was verified using LC-MS/MS-PRM assays targeting the Sulf-2 protein.

### 2.5. Spheroid Mono-Cultures and Co-Cultures

Tumor cell monolayers were washed with phosphate-buffered saline and recovered using cell dissociation enzyme, and the cell suspensions were centrifuged at 500× *g* for 5 min. The cell pellet was resuspended in 1 mL of complete growth medium and diluted to 6 × 10^4^ cells/mL (optimal cell density to obtain SCC35 and Cal33 spheroids of 300–500 µm diameter 24 h after cell seeding). From the cell suspension, 100 µL was dispensed per well into ultra-low attachment 96-well round bottom plates, and the cells were grown for 1 day in an incubator. The wells were visually inspected for tumor spheroid formation before proceeding with the 3D invasion assay.

For co-culture spheroids, an equal number of HNSCC and HNCAF37 cells (3 × 10^4^ cells/mL for each cell type) were mixed and seeded as described above for mono-culture spheroids.

### 2.6. Quantification and Inhibition of Sulf-2 Activity

We used a previously described fucosylated chondroitin sulfate from the sea cucumber *Holothuria floridana* (HfFucCS) to inhibit the activity of Sulf-2 [35,36]. The inhibitory potential of HfFucCS was determined using two assays. We adopted an arylsulfatase assay using 4-MUS [34] to a 384-well format as follows: 50 ng of Sulf-2 was incubated with 4 mM 4-MUS in 50 mM Tris, pH 7.5, 10 mM CaCl_2_, and 0.01% Tween 20 in a total volume of 25 µL in 384-well black plates for 4 h at 37 °C. The reaction was stopped by addition of 25 µL of 1 M Na_2_CO_3_, pH 11, and the fluorescence was measured at 360 nm (ex)/450 nm (em) using a GloMax Explorer plate reader (Promega, Madison, WI, USA). The heparan sulfatase activity was confirmed using our recently described 6-O desulfation assay using GlcNS6S-GlcA-Glc6SNS-IdoA2S-GlcNS6S-IdoA2S-GlcNS6S-GlcA-pNP (2S2-6S4) [34] with slight modifications. Briefly, the reaction consisted of 20 ng of Sulf-2, 100 µM 2S2-6S4 substrate, 50 mM Tris, pH 7.5, 10 mM CaCl_2_, and 0.01% Tween 20, and was carried out in 55 µL reaction volume for 4 h at 37 °C. Aliquots of 10 µL were collected at 0, 1, 2, 3, and 4 h. The substrate and the products of 6-O desulfation were separated using ion-exchange HPLC, detected at 310 nm, and quantified using automated peak integration with manual verification using OpenLab CDS Chemstation software (Rev.C.01.05.35) (Agilent, Santa Clara, CA, USA).

### 2.7. Disaccharide Analysis of Heparan Sulfate Desulfation

Twenty-five micrograms of heparan sulfate from porcine mucosa (Iduron, GAG-HS01) was digested with 1 µg of recombinant Sulf-2 [34] with or without 5 µg of HfFucCS in 50 mM Tris, pH 7.5, 10 mM CaCl_2_, and 0.01% Tween 20 in a 100 µL reaction at 37 °C for 8 h. The heparan sulfate was subsequently digested into disaccharide units using a combined treatment with heparinase I, II, and III and quantitatively analyzed using LC-MS/MS using 13C-labeleled disaccharide calibrants as described in detail previously [34,37].

### 2.8. HfFucCS Treatment of the Spheroid Cell Cultures

Once the spheroids formed, the plate containing the spheroids was chilled on ice for 10 min. Subsequently, 50 μL of Growth Factor Reduced Matrigel thawed on ice was added to each well. The plate was then spun at 100× *g* for 10 min and placed in a tissue culture incubator for 45 min to let the Matrigel polymerize. A total of 100 μL medium with/without HfFucCS (10 µg/mL) was gently added on top of the Matrigel in each well. Spheroids were cultured in an incubator and imaged using an Olympus IX71 inverted microscope (Olympus, Tokyo, Japan). Digital images of spheroids were analyzed using Particle Analysis in ImageJ software (NIH, Bethesda, MD, USA) to obtain the values of area and inverse circularity. Briefly, we used the sequence image acquisition → image type 8 bit → set threshold → analyze particles. Appropriate thresholding achieved fairly clean gating of the spheroids. In the analyze particles tool, we increased the lower limit of the size filter (Inch^2) to exclude all speckles and the results generated pertained only to the spheroids. The area for spheroids is the count of pixels comprising the object. Inverse *Circularity* = $\frac{1}{Circularity}$, where circularity (*Circularity* = 4π × (*area*)/(*perimeter*)) is a property of the spheroid calculated using automated image analysis with the analyze particles tool. The outline traces of the spheroids were analyzed to generate circularity measurements. A circularity value of 1 (maximum) indicates that the spheroid is perfectly circular. A decreasing value indicates less circular spheroids. The units are arbitrary units (A.U.) of measurement calibrated by the software. All our data processing and analysis parameters are identical for samples in the same dataset.

### 2.9. Targeted MS Analysis of the Sulf-2 Protein

Proteins from the SCC35, CAL33 (WT and KO), and HNCAF37 cell line conditioned media were digested using a Trypsin/Lys-C protease mix (Thermo, Waltham, WA, USA). The resulting peptides were fractionated and analyzed online using a nanoflow LC-MS/MS-PRM assay targeting the Sulf-2 peptide AEYQTAcEQLGQK (*m*/*z* 763.3512, *z* + 2).

### 2.10. Statistical Analysis

GraphPad Prism version 10 for Windows (GraphPad Software, La Jolla, CA, USA) was used for statistical analysis. Data are expressed as box and whisker plots, showing median and upper and lower quartile ranges. Differences in the area and inverse circularity of spheroids between control and test groups were evaluated using Student’s *t*-test. All statistical tests were based on a two-sided *p* value. Tests with *p* values < 0.05 were considered statistically significant unless specified otherwise.

## 3. Results

### 3.1. Generation of Sulf-2-Deficient HNSCC Cell Lines

We studied two HNSCC cell lines, SCC35 and Cal33, that express Sulf-2 to determine the impact of the enzyme on cell invasion into Matrigel. To this end, we used a recently reported SCC35 Sulf-2 knockout cell line (SCC35-KO), generated using CRISPR/Cas9 genome editing [34], and we created a Cal33-KO cell line using the same approach. The previously described SCC35 cells had an indel efficiency of 94% (R^2^ 0.87, knockout score 86) [34]; the newly generated Cal33 cells had an indel efficiency of 70% (R^2^ 0.81, knockout score 70) (Appendix A). The relative protein level of Sulf-2 in the secretome of these wild-type (WT) and respective -KO cell lines was determined using targeted mass spectrometry (Appendix A). Our LC-MS/MS-PRM assays corroborate the above knockout efficiencies; Sulf-2 protein was not detectable in the secretome of SCC35-KO, and a largely reduced expression was detected in the Cal33-KO secretomes (17% compared to the Cal33-WT). At the same time, we found that the secretome of HNCAF37, a primary HNSCC CAF, contains the Sulf-2 protein but in a lower amount (approximately half) compared to the cancer cell lines.

### 3.2. HNCAF37 Supports Invasion of HNSCC Cell Lines into Matrigel

We studied the invasive properties of the cancer cells in a spheroid culture model. We generated Matrigel-embedded mono-culture (HNSCC cells only) and co-culture (HNSCC + HNCAF37 cells) spheroids. Both SCC35 and Cal33 HNSCC cell lines formed spheroids in mono- and co-cultures on day 1 (Figure 1 and Figure 2). The co-culture spheroids clearly invaded the Matrigel during a 5-day experiment. On day 5, we observed an average 3-fold increase in size (area) and a 1.5-fold increase in inverse circularity (invasiveness) for SCC35 (Figure 1B), and a 4.5-fold increase in size (area) and a 2.3-fold increase in inverse circularity for Cal33 (Figure 2B). We assessed invasiveness using measurement of the irregularity of the spheroid surface determined as inverse circularity. Mono-culture spheroids, on the same timescale, showed marginal growth and invasion with a 1.2-fold increase in the area and a 0.9-fold change in inverse circularity (Figure 1A for SCC35 and Figure 2A for Cal33). Comparison of the mono- and co-culture spheroids on day 5 showed that HNCAF37-activated HNSCC cells generate cellular protrusions and invade Matrigel in a form resembling invasion of cancer cells into the collagen matrix. The co-culture spheroids were significantly larger in both the area and inverse circularity. The area increased on average 3-fold for SCC35 (Figure 1C) and 4-fold for Cal33 (Figure 2C). The inverse circularity increased on average 2.3-fold for SCC35 (Figure 1D) and 2.7-fold for Cal33 (Figure 2D).

### 3.3. HNCAF37-Mediated Invasion of HNSCC Cell Lines Depends on Sulf-2

We compared the invasion of HNSCC and their counterpart, Sulf-2-KO cells, in co-culture with HNCAF37. We observed that all the cells (HNSCC SCC35 and Cal33, Sulf-2-KO and HNCAF37) formed spheroids in mono-culture on day 1. The mono-culture spheroids retained their uniform circular form until day 5 (Appendix A). However, invasion of the spheroids occurred when the HNSCC or Sulf-2-KO cells were co-cultured with HNCAF37 (Figure 3A, SCC35; Figure 4A, Cal33). Interestingly, Sulf-2-KO co-culture showed significantly reduced invasion compared to the WT on day 5. The area of the co-culture spheroids was reduced, on average 0.6-fold, for SCC35KO (Figure 3B) and 0.5-fold for the Cal33KO (Figure 4B) compared to their WT counterparts. The WT SCC35 cells grew on average 3-fold on day 5 compared to day 1 (Figure 3); the Cal33 cells increased in size on average 4.5-fold under the same conditions (Figure 4). The Sulf-2-KO cells grew on average 1.8-fold for the SCC35-KO (Figure 3) and 2.1-fold for the Cal33-KO (Figure 4). Overall, the increase in the spheroid area on day 5 in the Sulf-2-KO cells was approximately 50% lower than in the WT cells.

The inverse circularity of the SCC35-KO cells was significantly reduced compared to the WT cells on day 5 (on average 5-fold, Figure 3C), and approximately 2-fold for the Cal33-KO (Figure 4C). The degree of change in the inverse circularity from day 1 to day 5 was different among WT and KO cells. We observed that SCC35 increased their inverse circularity on average 4-fold (Figure 3C) and Cal33 2.3-fold (Figure 4C), whereas the SCC35-KO did not change significantly (Figure 3C) and the Cal33-KO increased only 1.2-fold (Figure 4C). Thus, the invasiveness of the cells measured using the inverse circularity was decreased by the Sulf-2-KO in the two HNSCC cell lines. These results suggest that Sulf-2 regulates cancer cell invasion by adjusting as yet unidentified factors at the CAF–cancer cell interface.

### 3.4. HfFucCS Blocks Activity of Sulf-2 In Vitro

We searched for an effective inhibitor of the Sulf-2 activity that would confirm our Sulf-2-KO results but we did not find a good commercially available inhibitor. We therefore tested the inhibitory potential of a fucosylated chondroitin sulfate extracted from the holothurian species *H. floridana*, namely HfFucCS, using an arylsulfatase assay (4-MUS) and a heparan-specific 6-*O*-desulfation assay (2S2-6S4) that we developed recently [34]. The inhibition curves for both assays (Figure 5) showed similar characteristics with IC_50_ values of 0.14 and 0.18 ng HfFucCS per ng of enzyme for the 2S2-6S4 and 4-MUS assay, respectively. The HfFucCS (60–70 kDa) [36] is approximately three times smaller than the Sulf-2 enzyme (180–200 kDa). The calculated IC_50_ values therefore suggest that the complete inhibition of Sulf-2 is achieved at near equimolar concentrations of HfFucCS. We observed complete Sulf-2 inhibition at 1:20,000 (2S2-6S4) and 1:100,000 (4-MUS) inhibitor to substrate ratios. The HfFucCS is, therefore, a potent inhibitor but its mechanism of action is at this point unknown and needs to be further examined. The inhibition was further confirmed using a heparan sulfate disaccharide analysis, which showed that the major trisulfated disaccharide substrate of Sulf-2 (ΔUA2S-GlcNS6S) increased to nearly control level with the addition of HfFucCS while the major product (ΔUA2S-GlcNS) decreased (Appendix A).

### 3.5. HfFucCS Suppresses the Invasion of HNSCC Co-Culture Spheroids

The effect of the HfFucCS compound on invasion of cancer cells was monitored by treating HNSCC spheroids (WT HNSCC cells plus HNCAF37 co-cultures) on day 1 with a dose of 10 ng/µL. Compared to the no-treatment control (NTC), the HfFucCS-treated spheroids showed a significant reduction in invasion for both SCC35 (Figure 6) and Cal33 (Figure 7) cells. As expected, the SCC35 (Figure 6) and Cal33 (Figure 7) no-treatment control co-culture spheroids invaded Matrigel, but in the presence of HfFucCS the cellular protrusions were significantly reduced and the inverse circularity was on average 0.7-fold lower for SCC35 (Figure 6D) and 0.4-fold for Cal33 (Figure 7D) in the treated cells. The spheroid area was also significantly reduced; the size was on average 0.7-fold lower for SCC35 (Figure 6C) and 0.5-fold for Cal33 (Figure 7C) for the treated cells.

## 4. Discussion

HSPGs are essential components of the cell surfaces and ECM of normal tissues. Sulfation of the HSPGs is a critical determinant of protein binding [38,39] and a major regulatory system of cell–cell and cell–matrix interactions [40,41,42,43,44]. Protein–HSPG interactions reach nM dissociation constants [44,45] and the binding of many proteins to HSPGs is specifically sensitive to 6-*O*-sulfation which is regulated by the activity of the extracellular heparan-6-*O*-endosulfatases Sulf-1 and Sulf-2 [46]. Sulfation of the HSPGs is exquisitely regulated in normal development [41,43] but is perturbed in the microenvironment of tumors with profound oncogenic implications [20,26,27,47]. Receptor tyrosine kinase/ligand interactions, including FGF/FGFR, VEGF/VEGFR, HB-EGF/EGFR, PDGF/PDGFR, and WNT/Frizzled, are perhaps the best-established examples of HS-regulated oncogenic pathways [31,48]. However, HSPGs also distribute cytokines/chemokines, which adjust the immune responses [11,49], define the structure of the ECM [50], and regulate matrix remodeling [44].

Sulf-1 [27,31] and Sulf-2 [31,47] are upregulated and oncogenic in multiple cancers [20,29,47,51], as we summarized recently [28]. Sulf-1 is typically not expressed in cancer cell lines and its overexpression may suppress cell growth [52]. However, Sulf-1 is commonly overexpressed in tumors [28]. We have documented [30] that cancer-associated fibroblasts (CAF) supply Sulf-1 to HNSC tumors [53], which explains the cancer cell–tumor discrepancy. At the same time, high Sulf-1 in HNSCC and other cancers is associated with poor survival outcomes [28]. Sulf-2 is expressed primarily by epithelial cancer cells, is oncogenic in many cancers, and is associated with poor survival outcomes [28]. We have shown that Sulf-2 is upregulated and associated with poor survival outcomes in HNSCC [29,30]. We therefore focused our study on the CAF-assisted invasiveness of the cancer cells, observed by us and others [32,54,55], and its dependence on the Sulf-2 enzyme.

We have shown that Sulf-2 impacts the invasion of primary HNSCC CAF–cancer cell co-culture spheroids in the SCC35 and Cal33 HNSCC cell lines. This suggests that the secreted Sulf-2 enzyme remodels the cell–cell and cell–matrix interactions in as yet undefined ways that affect the invasion of the tumor cells into the matrix. We find that primary HNCAF37 is required for the SCC35 and Cal33 cells to invade Matrigel (Figure 1 and Figure 2). The cancer cell lines and CAF are all sources of the secretory Sulf-2 protein but the cancer cells contribute the majority of the enzyme (Appendix A). We thus created Sulf-2 knockout cell lines (SCC35-KO; Cal33-KO) using CRISPR-Cas technology as described [34], and we have shown that the Sulf-2-KO lines have a significantly decreased ability to invade Matrigel compared to the WT lines (Figure 3 and Figure 4). This supports our hypothesis that Sulf-2 enzymatic activity promotes local invasion of the cancer cells. The primary HNCAF37 cells are necessary for the cancer cell invasion but Sulf-2-KO in the cancer cells is sufficient to reduce the ability of the SCC35 and Cal33 cell lines to generate invadopodia and invade Matrigel.

Our results suggest that targeting Sulf-2 in HNSCC might be an interesting strategy to reduce local cancer cell invasiveness. Arresting cells in their early invasion into the matrix is expected to limit subsequent lymph node metastasis and could affect cancer outcomes. We therefore tested the ability of a fucosylated chondroitin sulfate GAG isolated from the sea cucumber species *H. floridana*, named HfFucCS [36], to inhibit Sulf-2 and cancer cell invasion. We chose this because a recent study has shown that a chondroitin sulfate modification negatively regulates the activity of the Sulf-2 enzyme [56], marine organisms are a rich natural source of unique GAG polymers [57], and efficient inhibitors of the Sulf-2 enzyme are not available. Several compounds were tested as Sulf-2 inhibitors [58,59,60] but they are either inefficient or not available. Fucosylated chondroitin sulfate is a unique GAG found primarily in sea cucumbers. This marine sulfated glycan is composed of a chondroitin sulfate backbone decorated with fucosyl branches attached to the glucuronic acid. The fucosylated chondroitin sulfates exhibit biological actions including inhibition of blood clotting and inhibition of severe acute respiratory syndrome coronavirus infection. These biological effects depend on the molecular weight (MW), number of fucosyl branches, and sulfation of both the branching as well as the backbone residues [35,36]. What specific structural features define the inhibition of the Sulf-2 enzyme by HfFucCS is as yet unknown and requires further study.

We show for the first time that HfFucCS is an effective inhibitor of Sulf-2 (Figure 5) using two different desulfation activity assays [34]. Subsequently, we evaluated the impact of the inhibitor on cancer cell invasion in our CAF–cancer cell spheroid model. Treatment of the co-cultures with HfFucCS reduced the invasion of both SCC35 and Cal33 cancer cell lines (Figure 6 and Figure 7). The HfFucCS could have additional mechanisms of action [36] but we show clearly that it potently inhibits the Sulf-2 enzyme and blocks cancer invasion in a manner similar to the Sulf-2 knockout in the cancer cells. The spheroid co-cultures model the cell–cell and cell–matrix interaction in the tumor microenvironment. Our results show that the invasiveness of HNSCC cells in this model depends on the Sulf-2 and warrant further exploration in vivo.

## 5. Conclusions

Overall, our study shows that the Sulf-2 enzyme affects the crosstalk of CAF with HNSCC cells and the invasion of cancer cells into Matrigel. We identified HfFucCS, a fucosylated and sulfated GAG polysaccharide, as an inhibitor of the Sulf-2 enzyme and of the CAF-assisted cancer cell invasion. Our results warrant further exploration of the heparan-6-*O*-endosulfatases and their inhibition in HNSCC and other cancers.

## Figures and Tables

**Figure 1 cancers-15-05168-f001:**
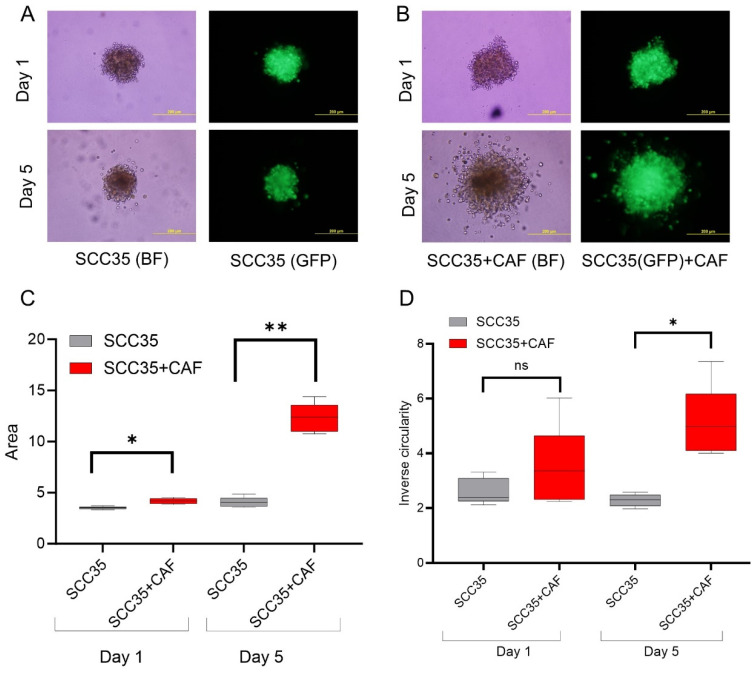
Primary head and neck cancer-associated fibroblast HNCAF37 induces invasion of SCC35 cells into Matrigel. (**A**) Representative images of spheroids of SCC35 cells in Matrigel on day 1 and day 5. (**B**) Representative images of co-culture spheroids of SCC35 with HNCAF37 in Matrigel on day 1 and day 5. (**C**) Changes in the area of the ‘mono-culture’ and ‘co-culture’ spheroids on day 1 and day 5 (optimum growth). (**D**) Changes in cellular invasion measured as inverse circularity on day 1 and day 5. Scale bars for panels (**A**,**B**) are 200 µm; images are presented as bright field (BF) or green fluorescence (GFP) for the labeled SCC35 cells. For graphs, red boxes represent SCC35 + CAF spheroids, grey boxes represent SCC35 spheroids. Graph whiskers represent the minimum and maximum, boxes extend from the 25th to the 75th percentile, and lines represent the median; n = 5 independent measurements of the spheroid area and the inverse circularity. Statistical significance was assessed using Student’s *t*-test comparing the mono-culture and co-culture spheroids. * indicates *p* ≤ 0.001, ** indicates *p* ≤ 0.0001, ns indicates no significant difference.

**Figure 2 cancers-15-05168-f002:**
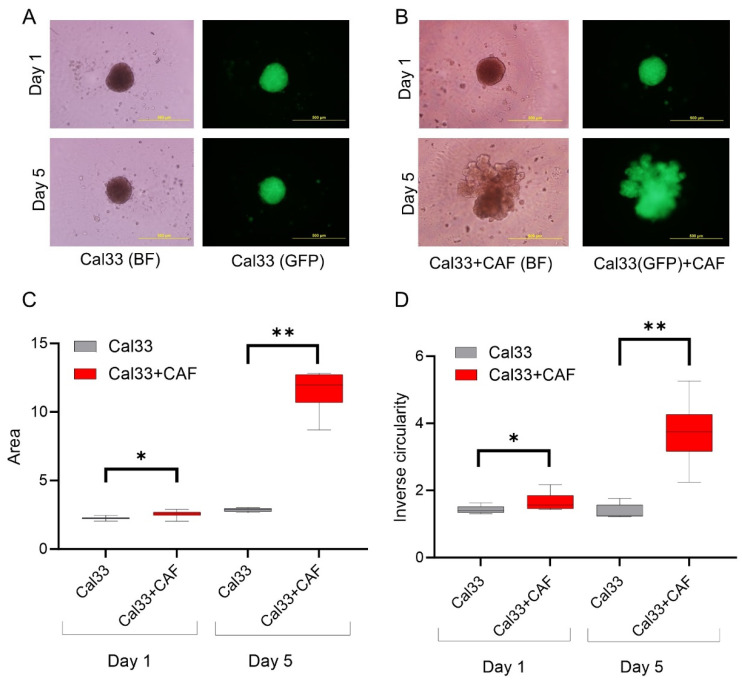
Primary head and neck cancer-associated fibroblast HNCAF37 induces invasion of Cal33 cells into Matrigel. (**A**) Representative images of spheroids of Cal33 cells in Matrigel on day 1 and day 5. (**B**) Representative images of co-culture spheroids of Cal33 cells with HNCAF37 in Matrigel on day 1 and day 5. (**C**) Changes in the area of the mono-culture and co-culture spheroids on day 1 and day 5. (**D**) Changes in the cellular invasion (inverse circularity) of the mono-culture and co-culture spheroids on day 1 and day 5. Scale bars for panels (**A**,**B**) are 500 µm; images are presented as bright field (BF) or green fluorescence (GFP) for the labeled Cal33 cells. For graphs, red boxes represent Cal33 + CAF spheroids, grey boxes represent Cal33 spheroids. Graph whiskers represent the minimum and maximum, boxes extend from the 25th to the 75th percentile, and lines represent the median; n = 5 independent measurements of the spheroid area and the inverse circularity. Statistical significance was assessed using Student’s *t*-test comparing the mono-culture and co-culture spheroids; * *p* ≤ 0.01, ** *p* ≤ 0.001.

**Figure 3 cancers-15-05168-f003:**
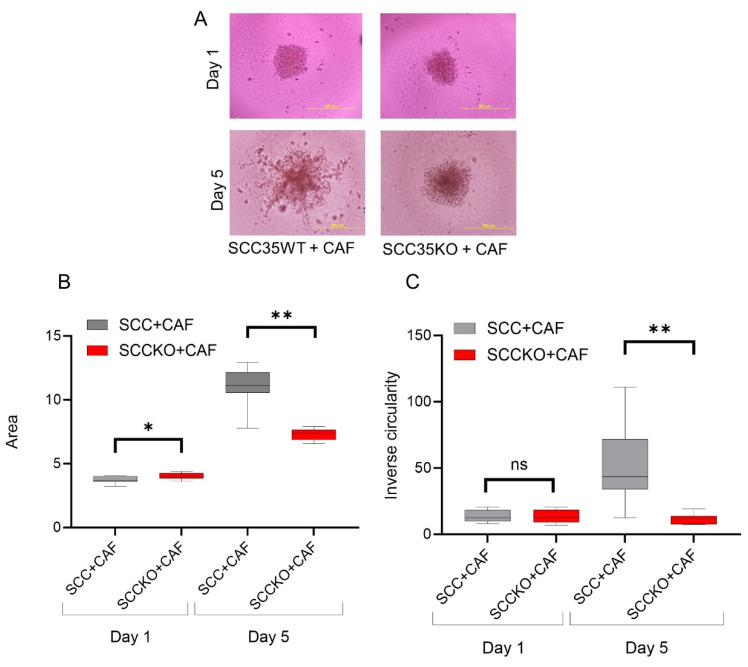
HNCAF37-driven invasion of SCC35 cells is reduced by Sulf-2 KO. (**A**) Representative images of co-culture spheroids of SCC35WT (wild type) and SCC35KO (Sulf-2-KO) cells with HNCAF37 on day 1 and day 5. (**B**) The SCC35WT and SCC35KO co-culture spheroid areas on day 1 and day 5. (**C**) Inverse circularity of the SCC35WT and SCC35KO spheroids on day 1 and day 5. Scale bars are 500 µm. For graphs, red boxes represent SCC35KO + CAF spheroids, grey boxes represent SCC35WT + CAF spheroids. Graph whiskers represent the minimum and maximum, boxes extend from the 25th to the 75th percentile, and lines represent the median; n = 5 independent measurements of the spheroid area and the inverse circularity. Statistical significance was assessed using Student’s *t*-test; * *p* ≤ 0.05, ** *p* ≤ 0.001, ns indicates no significant difference.

**Figure 4 cancers-15-05168-f004:**
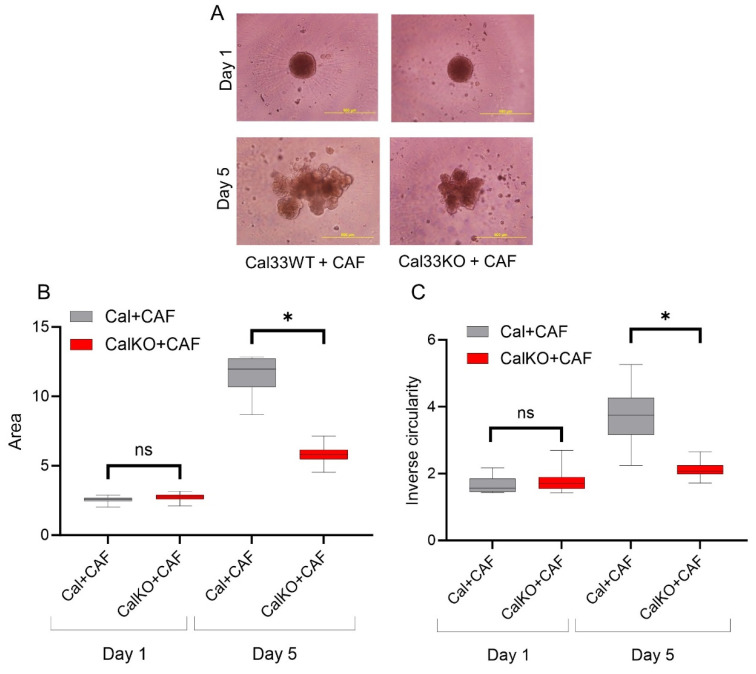
HNCAF37-driven invasion of Cal33 cells is reduced by Sulf-2 KO. (**A**) Representative images of co-culture spheroids of Cal33WT (wild type) and Cal33KO (Sulf-2-KO) cells with HNCAF37 on day 1 and day 5. (**B**) Area of the Cal33WT and Cal33KO co-cultures with HNCAF37 on day 1 and day 5. (**C**) Inverse circularity of the Cal33WT and Cal33KO co-cultures with HNCAF37 on day 1 and day 5. Scale bars are 500 µm. For graphs, red boxes represent Cal33KO + CAF spheroids; grey boxes represent Cal33WT + CAF spheroids. Graph whiskers represent the minimum and maximum, boxes extend from the 25th to the 75th percentile, and lines represent the median; n = 5 independent measurements of the spheroid area and the inverse circularity. Statistical significance was assessed using Student’s *t*-test; * *p* ≤ 0.001, ns indicates no significant difference.

**Figure 5 cancers-15-05168-f005:**
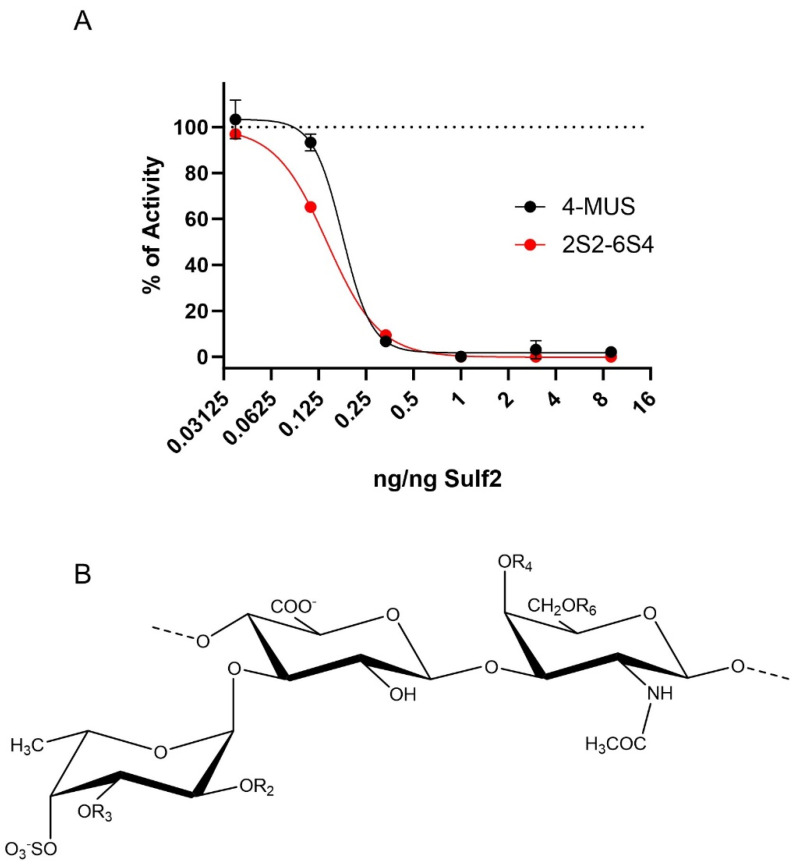
Effect of the marine fucosylated chondroitin sulfate HfFucCS on Sulf-2 activity. (**A**) Inhibition of Sulf-2 activity by HfFucCS measured using 4-MUS and 2S2-6S4 assays. Concentration of HfFucCS is shown as ng per ng of Sulf-2 enzyme. Results are expressed as percent activity of Sulf-2 without HfFucCS (dotted line). The IC_50_ values, obtained using the least squares fit, are 0.18 and 0.14 ng/ng of enzyme for the 4-MUS and 2S2-6S4 assay, respectively. (**B**) The structure of HfFucCS repeating unit having R2 = SO_3_^−^, R3 = H (45%); R2 = H, R3 = SO_3_^−^ (35%); R2 = R3 = H (20%); and R4 = R6 = SO_3_^−^ as the major form.

**Figure 6 cancers-15-05168-f006:**
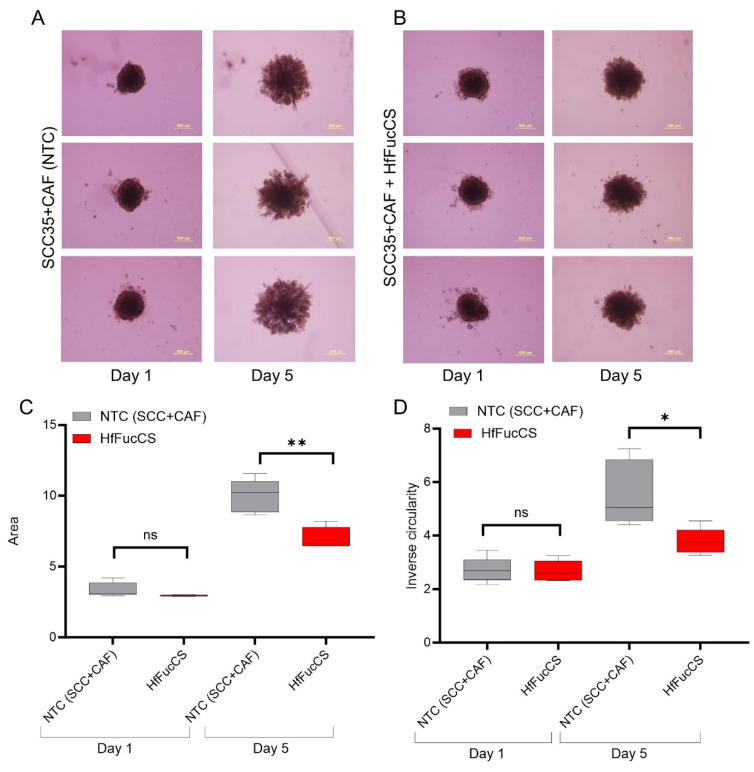
HfFucCS inhibits HNCAF37-mediated invasion of SCC35 cells. (**A**) Representative images of SCC35/HNCAF37 co-culture spheroids on day 1 and day 5; NTC, no-treatment control. (**B**) Representative images of HfFucCS-treated SCC35/HNCAF37 co-culture spheroids on day 1 and day 5. (**C**) Changes in the area of the co-culture spheroids with HfFucCS treatment measured on day 1 and day 5. (**D**) Changes in cellular invasion (inverse circularity) of the co-culture spheroids with HfFucCS treatment measured on day 1 and day 5. Scale bars are 500 µm. For graphs, red boxes represent HfFucCS-treated spheroids, grey boxes represent NTC spheroids. Graph whiskers represent the minimum and maximum, boxes extend from the 25th to the 75th percentile, and lines represent the median; n = 5 independent measurements of the spheroid area and the inverse circularity. Statistical significance was assessed using Student’s *t*-test; * indicates *p* ≤ 0.05, ** indicates *p* ≤ 0.01, ns indicates no significant difference.

**Figure 7 cancers-15-05168-f007:**
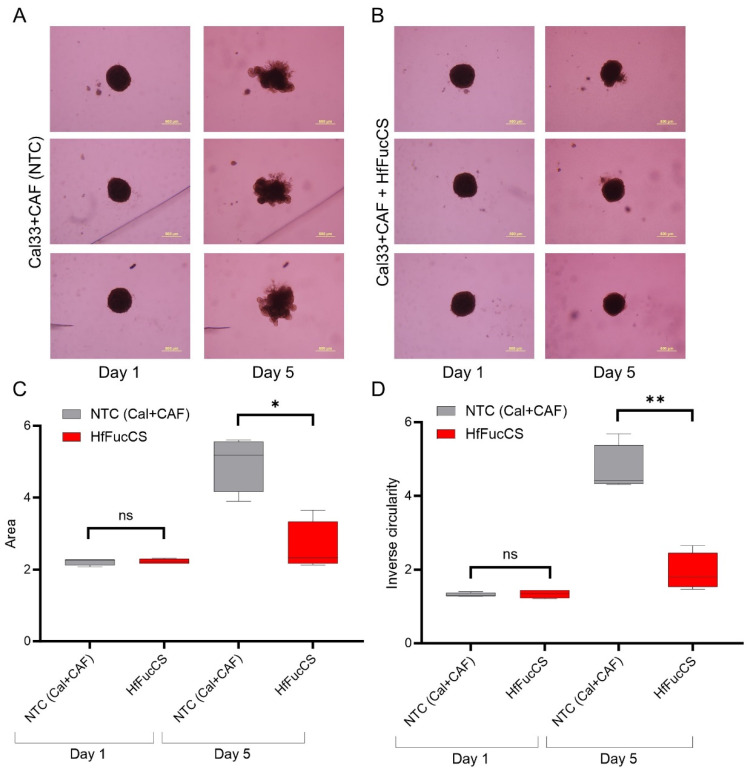
HfFucCS inhibits HNCAF37-mediated invasion of Cal33 cells. (**A**) Representative images of Cal33/HNCAF37 co-culture spheroids on day 1 and day 5; NTC, no-treatment control. (**B**) Representative images of HfFucCS-treated Cal33/HNCAF37 co-culture spheroids on day 1 and day 5. (**C**) Changes in the area of the co-culture spheroids with HfFucCS treatment measured on day 1 and day 5. (**D**) Changes in cellular invasion (inverse circularity) of the co-culture spheroids with HfFucCS treatment measured on day 1 and day 5. Scale bars are 500 µm. For graphs, red boxes represent HfFucCS-treated spheroids; grey boxes represent NTC spheroids. Graph whiskers represent the minimum and maximum, boxes extend from the 25th to the 75th percentile, and lines represent the median; n = 5 independent measurements of the spheroid area and the inverse circularity. Statistical significance was assessed using Student’s *t*-test; * indicates *p* ≤ 0.01, ** indicates *p* ≤ 0.001, ns indicates no significant difference.

## Data Availability

The data presented in this study are available in this article.

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
