# Peer review of "Heparan-6-O-Endosulfatase 2 Promotes Invasiveness of Head and Neck Squamous Carcinoma Cell Lines in Co-Cultures with Cancer-Associated Fibroblasts"

_cancers, 2023, doi:10.3390/cancers15215168_

Round 1

Reviewer 1 Report

Comments and Suggestions for Authors

The article by Mukherjee et al. is devoted to the study of the effect

of Heparan-6-O-endosulfatase 2 on the invasiveness of head and

neck squamous carcinoma cell lines in co-cultures with cancer

associated fibroblasts. Taking into account the fact that in

approximately 50% of cases of Head and neck cancer metastasis

to the lymph nodes is observed, the development of new diagnostic

and treatment methods to prevent the spread of the disease is an

urgent task.
The article may be of interest both to specialists in the field of Head

and neck cancer and to researchers working in the field of 3D

cultivation.
The material of the article is logically presented, the conclusions are

justified.
However, there are a number of questions and comments.
1. Could the authors duplicate the abbreviations introduced in the

Simple summary” to the abstract to facilitate understanding of the

material, given that the abstract is an annotation of the work.
2. Scale bars in pictures are hard to read. The figure legends say that

the scale bars correspond to 200 µM, however, if you zoom in on the

page, at least in figures 3, 4, 6, one can see that the scale bars

correspond to 500 µM.
3. Why are all the data given for such a small number of spheroids (n=5)?
4. How was the area of the spheroids calculated - was it the surface

area or the cross-sectional area of the maximum diameter? In what

units of measurement is area represented? This is not reflected on the axes.
5. Could the authors also describe in more detail how they calculated

inverse circularity and in what units of measurement the values are given.
6. Why in Fig. 1 and 2, in the case of mixed spheroids, all cells are

GFP-positive and the contribution of fibroblasts is not visible, although

the initial cell ratio was 1:1? Do HNCAF37 cells themselves grow in

Matrigel?
7. Why do the authors conclude that “the HfFucCS inhibitor arrests

growth... the HNSCC co-cultures” (line 364), if, at least for inhibitor-

treated SCC35/HNCAF37 spheroids, the area increased relative to 1

day (Fig. 6C )?
8. Have the authors tried to obtain mixed spheroids based on other

fibroblasts? Known human fibroblast cell lines or primary fibroblasts

from other origins? Can the authors speculate whether the patterns of

HNSC cell invasion would be maintained in this case?

Reviewer 2 Report

Comments and Suggestions for Authors

This is an interesting study about heparan-6-O-endosulfatase 2 nad its role in promoting invasiveness of head and neck squamous carcinoma cell lines in co-cultures with cancer associated fibroblasts. Two cell lines were used.

The authors generated Sulf-2 deficient HNSCC cell lines by gene editing using Crispr/Cas9, and their phenotypes were compared to wild-types. Furthermore, the effect of a novel inhibitor, a unique marine GAG named HfFucCS, on Sulf-295 activity was assessed. The study showed that Sulf-2 affected the growth and invasion of HNSCC spheroids and HfFucCS inhibited not only the Sulf-2 enzymatic activity but also cancer cell invasion in the spheroid model.

The paper is well written. However, some issues remain.

In the Introduction, please better specify the type of neoplasia referred to reference n.25.

Did the authors have any data after day 5?

Please add more information about molecular mechanisms of crosstalk between tumor cells and CAFs that may promote invasiveness.
